# Viterbi-based Pruning for Sparse Matrix with Fixed and High Index Compression Ratio

**Dongsoo Lee[1,2]\*, Daehyun Ahn[1], Taesu Kim[1], Pierce I. Chuang[3], Jae-Joon Kim[1]**
[1]POSTECH, Department of Creative IT Engineering, Korea
[2]Samsung Research, Korea
[3]IBM Thomas J. Watson Research Center, USA
`dongsoo3.lee@samsung.com,{daehyun.ahn,taesukim}@postech.ac.kr`
`pchuang@us.ibm.com, jaejoon@postech.ac.kr`

## Abstract

Weight pruning has proven to be an effective method of reducing the model size and computation cost without sacrificing its model accuracy. Conventional sparse matrix formats, however, involve irregular index structures with large storage requirement and a sequential reconstruction process, resulting in inefficient use of highly parallel computing resources. Hence, pruning is usually restricted to inference with a batch size of one, for which an efficient parallel matrix-vector multiplication method exists. In this paper, a new class of sparse matrix representation is proposed utilizing the Viterbi algorithm that has a high, and more importantly, fixed index compression ratio regardless of the pruning rate. In this approach, numerous sparse matrix candidates are first generated by the Viterbi encoder, and the candidate that aims to minimize the model accuracy degradation is then selected by the Viterbi algorithm. The model pruning process based on the proposed Viterbi encoder and Viterbi algorithm is highly parallelizable, and can be implemented efficiently in hardware to achieve low-energy and a high-performance index decoding process. Compared with the existing magnitude-based pruning methods, the index data storage requirement can be further compressed by 85.2% in MNIST and 83.9% in AlexNet while achieving a similar pruning rate. Even compared with the relative index compression technique, our method can still reduce the index storage requirement by 52.7% in MNIST and 35.5% in AlexNet.

## 1 Introduction

Deep neural networks (DNNs) demand an increasing number of parameters as the required complexity of tasks and supporting number of training data continue to grow (Bengio & Lecun, 2007). Correspondingly, DNN incurs a considerable number of computations and amount of memory footprint, and thus requires high performance parallel computing systems to meet the target response time. As an effort to realize energy-efficient DNN, researchers have suggested various low-cost hardware implementation techniques. Among them, pruning has been actively studied to reduce the redundant connections while not degrading the model accuracy. It has been shown that pruning can achieve $9\times$ to $13\times$ reduction in connections (Han et al., 2015).

After pruning, the remaining parameters are often stored in sparse matrix formats. Different ways of representing indices of non-zero values constitute the different sparse matrix format, and have a significant impact on the level of achievable computational parallelism when a sparse matrix is used as an input operand (Bell & Garland, 2009). If the format is not properly designed, then the performance of DNN with a sparse matrix can be even lower than the case with dense matrix (Yu et al., 2017). The two most important characteristics of a hardware-friendly sparse matrix format are 1) reducing index storage footprint and 2) parallelizable index decoding process. As a compromise between index size reduction and index decoding complexity, numerous formats have been proposed (Bell & Garland, 2009).

---

\*Work done while at POSTECH.

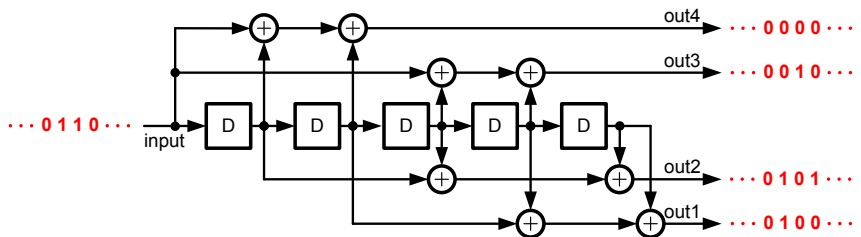

Figure 1: Viterbi decompressor (VD) structure.

$$\begin{bmatrix} 0 & 0 & 0 & 0 \\ 5 & 8 & 0 & 0 \\ 0 & 0 & 3 & 0 \\ 0 & 6 & 0 & 0 \end{bmatrix}$$

Dense Matrix after Pruning

A = [ 5 8 3 6 ]
IA = [ 0 0 2 3 4 ]
JA = [ 0 1 2 1 ]

CSR Format

A = [ 5 8 3 6 ]
I = [ 0 1 1 0 ]

VCM Format

Outputs of Viterbi decompressor
[ 0 0 0 0 ] 1st cycle
[ 1 1 0 0 ] 2nd cycle
[ 0 0 1 0 ] 3rd cycle
[ 0 1 0 0 ] 4th cycle

Figure 2: CSR Format and the proposed sparse matrix format comparison.

DNN after pruning heavily involves sparse matrix-vector and matrix-matrix multiplications (SpMV and SpMM, respectively). Despite the sparse content, the computation time for SpMM is longer than that of dense matrix multiplication in the modern graphic processing unit (GPU), due to its serialized index decoding process and irregular memory access patterns. For example, the inference latency of AlexNet and VGG16 with SpMM can be increased by $2\times$ to $5\times$ on GPUs or CPUs (Han et al., 2016a). The traditional pruning technique, therefore, is only attractive in the case where SpMV can be utilized (i.e., batch size of 1) (Han et al., 2016b) (Yu et al., 2017). Therefore, a sparse matrix representation associated with parallelizable dense-matrix reconstruction in a wide range of computing operations is the key to extending the use of pruning.

We propose a new DNN-dedicated sparse matrix format and a new pruning method based on error-correction coding (ECC) techniques. A unique characteristic of this sparse matrix format is the fixed, yet high (as shown in Section 3) index compression ratio, regardless of the pruning rate. Moreover, sparse-to-dense matrix conversion employing the proposed format becomes a parallel process and is no longer the performance bottleneck. Notice that conventional sparse matrix formats entail at least one column or row index value for each non-zero parameter such that the amount of index data is larger than that of non-zero values. On the other hand, the proposed approach compresses the locations of non-zero values with a convolutional code which is a type of ECC code. Consequently, the size of the sparse matrix index becomes negligible.

Conventional pruning approaches first identify the parameter candidates to be pruned, then construct a matrix (often sparse) using formats such as Compressed Sparse Row (CSR) to represent the survived parameters. On the contrary, in the proposed scheme, pruning is performed in a restricted manner since a specific sparse matrix format is first constructed. A DNN-specific Viterbi encoder takes an input pattern and generates a sequence of random-number, where a "1" indicates the parameter had survived, and had been pruned otherwise. Depending on the length of the input pattern, a vast (but limited) number of output patterns (hence candidates of the final sparse matrix representations) are considered. In this case, the input pattern is used as the sparse matrix index. The content of the input pattern, which generates a deterministic output random number sequence, is chosen such that the accuracy degradation is minimized based on a user-defined cost function (more details on Section 2). Both the Viterbi encoder and the algorithm have been shown to be computationally efficient with an inherent parallelizable characteristic, as demonstrated in the digital communication applications (Viterbi, 1998). In this work, we further extend its application and demonstrate how the Viterbi algorithm can be modified to perform energy-efficient DNN pruning.

## 2 PRUNING USING VITERBI-BASED APPROACH

Figure 1 illustrates the proposed Viterbi decompressor (VD), which is based on the Viterbi encoder widely used in digital communication. VD has a simple structure consisting only of FlipFlops (FFs) and XOR gates. In this configuration, VD takes one input bit and produces four output bits every clock cycle. Notice that FFs and XOR gates intermingle input bits and generate pseudo random number outputs. Assume that a dense matrix is formed after pruning, as shown in Figure 2, and an input sequence of $\{0, 1, 1, 0\}$ is applied to VD through four clock cycles to generate the outputs, where '1' implies that the corresponding parameter has survived. In this case, the overhead in the index for the proposed Viterbi-Compressible Matrix (VCM) format is significantly less than that of CSR. In the VCM format, the input sequence to the VD becomes the index information. This index size is independent of the number of non-zero values and can be determined in advance based on the target index compression ratio[1]. Unlike the CSR format, the available VD-compressible dense matrix representation is limited, meaning that not all possible dense matrix representations after conventional magnitude-based pruning (such as (Han et al., 2015)) can be reconstructed by VD. Therefore, the pruning method considering VCM may result in a matrix that contains different survived parameters compared to a pruning method using the CSR format. Thus, the key to the success of VCM is to design a VD that allows diversified parameters to survive, and to efficiently search for the optimal VD input sequence that minimizes the accuracy degradation[2].

### 2.1 VITERBI DECOMPRESSOR (VD) DESIGN CONSIDERATIONS

If the input sequence length and the total output sequence length of a VD are denoted as $p$ and $q$, respectively, then the index compression ratio can be calculated as $q/p$. Achieving a high index compression ratio (i.e., $q >> p$) implies that the possible $2^p$ VD-compressible dense matrix representations need to be uniformly distributed inside the $2^q$ space to maximize the likelihood of finding a dense matrix representation that is closely matched to the optimal case.

In other words, the goal of VD is to act as a random number generator using the input sequence. It is interesting to note that such an effort has already been studied in ECC design (Morelos-Zaragoza, 2006). Since "random coding" has been introduced by C. Shannon to prove his channel capacity model (Shannon, 1948), practical ECC techniques with a fixed encoding rate was proposed to simulate random coding with an allowed decoding complexity. We choose the Viterbi encoder, which is the base model of VD, as a controllable random number generator because of its simplicity and flexible design when increasing the number of outputs. The randomness of VD outputs is determined by the number of FFs and the XOR gates configuration. We present the details of the VD design methodology in Appendix A.1.

The basic structure of VD is similar to the design introduced in (Lee & Roy, 2012). VD targeting DNN applications, however, requires the number and/or distribution of 1 (i.e., pruning rate) to be a user-defined parameter, whereas in the typical applications that require random number generation, such as ECC and VLSI testing, the number of 1s and 0s should be approximately the same. In order to control the pruning rate, the VD outputs are connected to binary number comparators. For instance, in Figure 1, one input of the comparator takes a two-bit number $\{out2, out1\}$, while the other input takes a user-defined threshold value ($TH_c$). If $\{out2, out1\}$ (or $\{out4, out3\}$) is larger than $TH_c$, the comparator produces a "1", and a "0" otherwise. A trade-off occurs between the granularity of the pruning rate and the index compression ratio. If the number of VD outputs, the number of comparator input bits, and the number of comparators (i.e., index compression ratio) are denoted as $NUM_v$, $NUM_c$, and $R$, respectively, then $NUM_v = NUM_c \times R$ (see Figure 10). The proposed index decoding operation utilizing VD is inherently a parallel process with a small hardware overhead. Unlike CSR or other similar formats that employ an irregular index structure, decoding VCM using VD does not incur significant buffer/memory overhead for indices and/or non-zero values, and most importantly, can be performed with a fixed and predictable rate. A fixed index compression ratio is also desirable for efficient memory bandwidth utilization and for applying the tiling technique to further improve the level of parallelism.

---

[1]As an example, the structure shown in Figure 1 provides four output bits per one input bit, achieving an index compression ratio of four

[2]In the context of magnitude-based pruning, the objective of pruning using the VD is to identify a set of VD input sequences that preserves maximum number of larger value weights

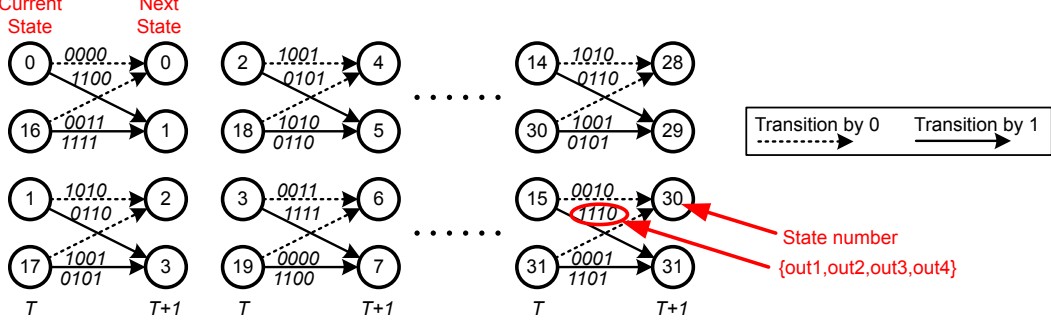

Figure 3: Trellis diagram of VD shown in Figure 1.

## 2.2 Viterbi Algorithm for Pruning

The basic idea of our proposed pruning method is to assign a cost function to each pruning case enabled by VD and evaluate all possible ($2^p$) pruning cases with a "Branch-and-Cut" algorithm. The pruning case that has the optimal (i.e., lowest) cost function should lead to minimal accuracy degradation. The Viterbi algorithm computes the maximum-likelihood sequence in a hidden Markov model (Forney, 1973), and can be utilized as a fast and efficient pruning exploration technique for our pruning method. Pruning using Viterbi algorithm follows the next 3 steps.

The first step involves constructing a trellis diagram which is a time-indexed version of a state diagram. A state of VD can be represented using FF values, where the leftmost FF value becomes the least significant bit. If VD has $k$ FFs, the total number of states is $2^k$. Hence, VD in Figure 1 has a total of 32 states as shown in Figure 3, where $T$ is the time index. Each possible transition with an input bit (0 or 1) produces multiple corresponding output bits. A trellis diagram holds the entire operations inside VD in a compact fashion.

The next step involves computing a cost function for possible transitions using the branch metric and the path metric. The branch metric is expressed as $\lambda_t^{i,j}$ where $t$ is a time index and $i$ is a predecessor state of $j$. $\lambda_t^{i,j}$ denotes the cost of traversing along a transition from $i$ to $j$ at the time index $t$. By accumulating the branch metrics and selecting one of two possible transitions reaching the same state at the same time index, the path metric is defined as

$$\Gamma_{t+1}^j = \max\left(\Gamma_t^{i1} + \lambda_t^{i1,j}, \Gamma_t^{i2} + \lambda_t^{i2,j}\right), \tag{1}$$

where $i1$ and $i2$ are two predecessor states of $j$. In practice, path metrics can be normalized to avoid overflow. Note that we use $\max$ function for the path metric instead of $\min$ function in Eq. (1) because the metric values in our method describe a degree of 'reward' rather than 'cost'. For the entire "survived path" selections during the path metric update, the decisions are stored in the memory and the old path metrics can be discarded. The objective of this Viterbi algorithm is to find a path maximizing the accumulation of the branch metrics ($\lambda_t^{i,j}$), which is expressed as:

$$D_t^{i,j,m} = \left(W_t^{i,j,m} - TH_p\right)/S_1, \ 0 \le W_t^{i,j,m}, TH_p \le 1$$

$$\beta_t^{i,j,m} = \begin{cases} \tanh\left(D_t^{i,j}\right) \times S_2, & \text{when survived} \\ -\tanh\left(D_t^{i,j}\right) \times S_2, & \text{when pruned} \end{cases}, \ \lambda_t^{i,j} = \sum_{m=1}^{R} \beta_t^{i,j,m}, \tag{2}$$

where $W_t^{i,j,m}$ is the magnitude of a parameter at the $m^{\text{th}}$ comparator output and time index $t$, normalized by the maximum magnitude of all parameters inside the dense matrix to be pruned, and $TH_p$ is the pruning threshold value. Intuitively, $\beta_t^{i,j,m}$ favors(discourages) the survival(pruning) of parameters with larger magnitude through the skewed $\tanh$ function. Pruning with the Viterbi algorithm is flexible such that different cost function can be assigned to the branch metric, depending on the type of pruning approach, providing the pruning algorithm follows a hidden Markov model (Lou, 1995)[3]. The two constants, $S_1$ and $S_2$, are the scaling factors, and are empirically determined to be

---

[3]Eq. (2) in this work is related to magnitude-based pruning

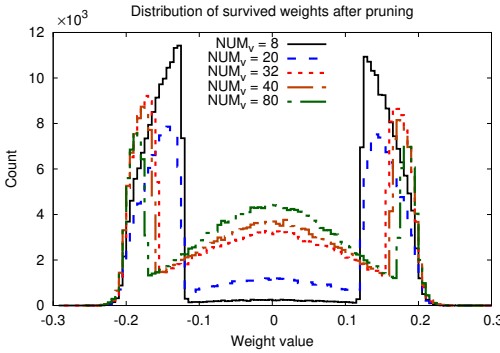

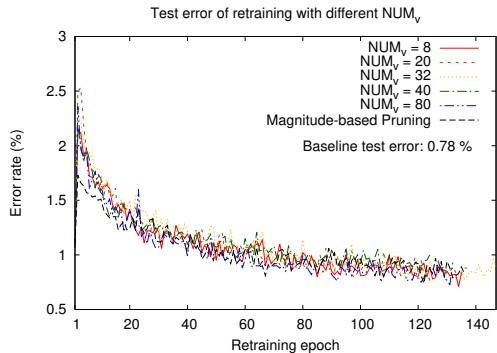

Figure 4: Distribution of the weights of FC1 after pruning with different $NUM_v$.

Figure 5: Test error of retraining with different $NUM_v$.

5.0 and $10^4$, respectively, for our experiments. Note that exploring diversified states (and hence, various pruning cases) is achieved by maintaining approximately $50\%$ of the '1' and '0' distributions for both inputs and outputs of VD (Forney, 1973). Consequently, the target pruning rate is mainly controlled by the comparator threshold value, $TH_c$ (e.g., if $TH_c$ is a 4-bit number and $TH_c$=3, then $25\%(= (3 + 1)/2^4)$ is the target pruning rate). $TH_p$ is determined by considering the distribution of parameters and the given target pruning rate (e.g., if the parameters follow a Gaussian distribution and the target pruning rate is $68.3\%$, $TH_p$ corresponding to one sigma is recommended).

Once the final time index is reached, as the last step of Viterbi pruning, the state with the maximum path metric is chosen, and the previous state is traced by reading the surviving path selection data. We continue this trace-back procedure to the first time index of a trellis diagram. Note that if the initial states of FFs are all 0s, then the number of available states (hence the number of sparse matrix representations in the first few time indices) may be limited. As an alternative, a dummy input sequence having the length equal to the number of FFs[4] in VD can be inserted such that every state of VD is reachable (refer to Figure 11). In this case, the compressed input index of the VCM is a combination of the survived dummy sequence and the input sequence. It should be noted that the Viterbi algorithm can be implemented using a dynamic programming technique. The time complexity required to find the best pruning method becomes $O(l \cdot 2^f)$ where $l$ is the length of the input sequence and $f$ is the number of FFs. As can be seen in Appendix A.1, $f$ is small even with a large number of VD outputs.

# 3 EXPERIMENTAL RESULTS

In this section, the impact of different VD configurations and branch metric selections on model accuracy and the index compression ratio is analyzed. We empirically study the weight distribution after pruning and the sensitivity of accuracy using MNIST. Then, the observations from MNIST are applied to AlexNet to validate the scalability of our proposed method.

## 3.1 VD DESIGN AND BRANCH METRIC EXPLORATION USING MNIST

We perform experiments using the LeNet-5-like convolutional MNIST model[5]. For simplicity, both the minimum Hamming distance and the XOR taps (introduced in Appendix A.1) are fixed to be 4, and $NUM_c$ is 4 (i.e., $NUM_v = 4 \times R$). These parameters are selected for fast design exploration, and increasing them will enhance randomness of VD output and target pruning rate resolution which are critical to improving pruning rate with minimal accuracy degradation.

**Number of VD outputs ($NUM_v$):** Immediately after training, we prune the weights with different $NUM_v$ for VD. Figure 4 shows the weight distributions after pruning in the FC1 layer with fixed $TH_c$ and $TH_p$. Lower $NUM_v$ (i.e, lower index compression ratio) leads to sharper pruning around

---

[4]The storage overhead of this dummy input sequence is negligible compared to the index data storage

[5]https://github.com/tensorflow/tensorflow/blob/r1.3/tensorflow/examples/tutorials/mnist/mnist_deep.py

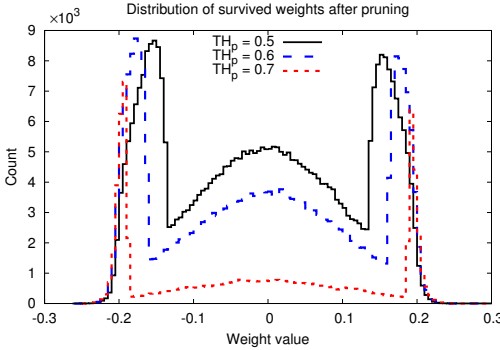

Figure 6: Distribution of FC1's weights after pruning with different $TH_p$.

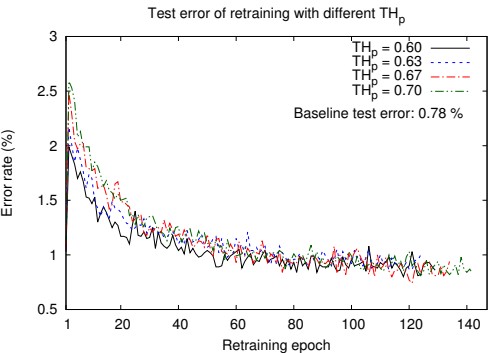

Figure 7: Test error of retraining with different $TH_p$.

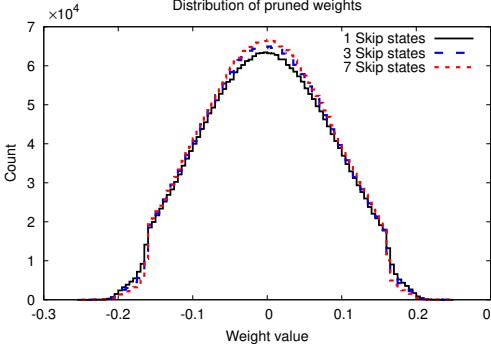

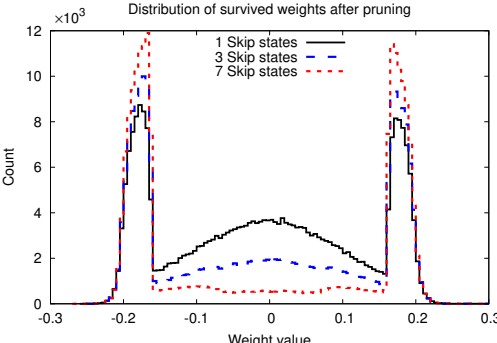

Figure 8: Distributions of pruned (Left) and survived (Right) FC1 weights with different skip state.

the weight determined by $TH_p$. Hence, $NUM_v$ provides a trade-off between accuracy and the index compression ratio. Extensive experiments indicate that for the Conv layer, a low $NUM_v$ is desired, while for the FC layer, a wide range of $NUM_v$ can lead to minimal accuracy degradation as shown in Figure 5 (magnitude-based pruning is from (Han et al., 2015)). For MNIST, $NUM_v$=8 for Conv layers and $NUM_v$=40 for FC layers have been chosen to achieve optimal trade-off between the index compression ratio and accuracy.

**Pruning threshold value ($TH_p$):** Even when the parameters before pruning follow a known distribution (e.g., Gaussian), it may still be an iterative task to search for an optimal $TH_p$ that results in the target pruning rate, especially with high $NUM_v$, as evident from Figure 4. Thus, it is necessary to investigate the sensitivity of accuracy to $TH_p$. In Figure 6, $TH_p$ affects distributions of survived weights and pruning rates given the same $TH_c$. Note that if the actual pruning rate differs from the target pruning rate, then VD outputs exhibit skewed supply of '1's or '0's to the comparators and the trellis diagram path exploration is also biased. In contrast, Figure 7 clearly shows that all the retraining processes converge, despite the minor discrepancy between the target and actual pruning rate (target pruning rate is 93.75%).

**Skip state** (Appendix A.2): Up to now, we have only considered the case where one input bit is supplied to VD at every clock cycle. However, if $n$ input bits are provided to VD at every clock cycle, then $n - 1$ time indices in a trellis diagram are skipped. While this results in a lower index compression ratio, which is defined as $R$ / (skip state + 1), the skip state allows for a more diverse state exploration and improves the pruning quality. As can be seen in Figure 8, a greater number of larger magnitude weights are preserved with increasing number of skip states while fixing both $TH_p$ and $NUM_v$. In this work, the default skip state is one.

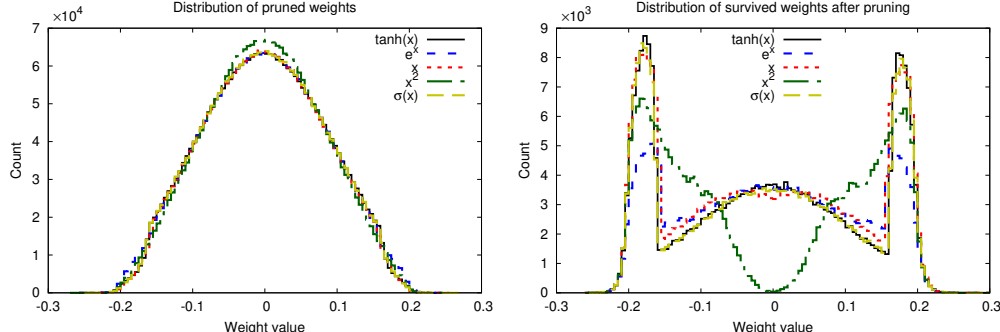

Figure 9: Left: Distribution of pruned (Left) and survived (Right) FC1 weights with different branch metric equations.

Table 1: MNIST test error and comparator threshold values with gradual pruning. Pruning is performed at the $50^{th}$ epoch ($\sim 50\%$ target pruning rate), $100^{th}$ epoch ($\sim 70\%$ target pruning rate), and $150^{th}$ epoch (final). 40 VD outputs are used for FC1, while 8 VD outputs for the others.

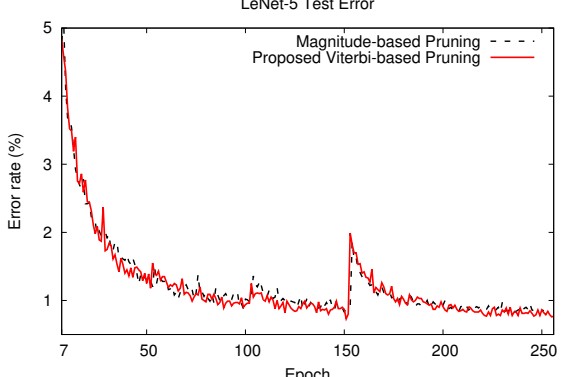

|       | comparator threshold value | | |
| Layer | $50^{th}$ Epoch | $100^{th}$ Epoch | $150^{th}$ Epoch |
|---|---|---|---|
| Conv1 | 4 | 4 | 4 |
| Conv2 | 7 | 10 | 12 |
| FC1 | 7 | 10 | 14 |
| FC2 | 7 | 10 | 12 |

Table 2: Sparse matrix comparison with MNIST using magnitude-based pruning (Han et al., 2015) and our proposed Viterbi-based pruning. We assume that 16 bits are used for the non-zero values and index for magnitude-based pruning.

|       | Weight Size | Magnitude-Based | | Viterbi-Based | | Matrix Size Reduction |
| Layer | | Pruning Rate | Sparse Matrix (CSR) Size | Pruning Rate | Sparse Matrix (VCM) Size | |
|---|---|---|---|---|---|---|
| Conv1 | 0.8K | 34.4% | 2.12KB | 32.3% | 1.16KB | 45.3% |
| Conv2 | 51.2K | 87.4% | 25.41KB | 81.3% | 24.98KB | 1.7% |
| FC1 | 3211.3K | 91.0% | 1125.54KB | 93.1% | 512.82KB | 54.4% |
| FC2 | 10.2K | 81.1% | 7.62KB | 80.4% | 5.17KB | 32.2% |
| Total | 3273.5K | **90.9%** | 1160.69KB | **92.8%** | 544.13KB | **53.1%** |
| Test Error | | 0.77% | | 0.78% | | |

**Branch Metric**: For the branch metric, a variety of functions, such as $e^x$ and the sigmoid function $\sigma(x)$, has been investigated, as shown in Figure 9. Among them, the "tanh" function is selected due to its pruning sharpness and low sensitivity to $TH_p$ and $NUM_v$.

Based on the observations discussed above, we conducted a pruning and retraining process, and compared the test errors of the magnitude-based pruning method (Han et al., 2015) and the proposed Viterbi-based pruning method. For every round of pruning, all the weights, including the ones pruned in the previous run, are considered. Table 1 illustrates the comparator threshold values $TH_c$

Table 3: Pruning and sparse matrix size comparison for AlexNet on ImageNet using magnitude-based pruning (Han et al., 2015) and our proposed Viterbi-based pruning. We assume that 16 bits are used for the non-zero values and index for magnitude-based pruning.

| Layer | Weight Size | Magnitude-Based | | Viterbi-Based | | Matrix Size Reduction |
|---|---|---|---|---|---|---|
| | | Pruning Rate | Sparse Matrix (CSR) Size | Pruning Rate | Sparse Matrix (VCM) Size | |
| Conv1 | 34.8K | 16% | 69.70KB* | - | 69.70KB* | 0.0% |
| Conv2 | 307.2K | 62% | 467.46KB | 62.5% | 268.99KB | 42.5% |
| Conv3 | 884.7K | 65% | 1239.40KB | 62.3% | 777.21KB | 37.3% |
| Conv4 | 663.6K | 63% | 982.82KB | 62.0% | 586.73KB | 40.3% |
| Conv5 | 442.4K | 63% | 655.22KB | 56.0% | 444.83KB | 32.1% |
| FC1 | 37.7M | 91% | 13597.74KB | 90.3% | 8284.93KB | 39.1% |
| FC2 | 16.8M | 91% | 6047.99KB | 90.8% | 3505.43KB | 42.0% |
| FC3 | 4.1M | 75% | 4098.00KB | 73.7% | 2670.18KB | 34.8% |
| Total | 61.0M | **89%** | 27158.31KB | **88.2%** | 16607.99KB | **38.1%** |
| Test Error (Top-1) | | 42.73% | | 42.68% | | |
| Test Error (Top-5) | | 19.77% | | 19.78% | | |

*Dense matrix size is considered in this layer because both CSR and VCM representation result in a larger memory footprint due to the low pruning rate.

(MIN=0, MAX=15 with $NUM_c$=4) used for each pruning round and test error results. Since Conv1 is close to the input nodes, we choose a smaller $TH_c$ to reduce the target pruning rate of Conv1. From Table 1, it is clear that the proposed pruning method successfully maintains accuracy during the entire training process.

The final pruning rate and memory requirement for CSR and VCM for each layer are summarized in Table 2. Notice that the sparse matrix represented using the VCM format leads to a significant memory footprint reduction (by **53.1%**) compared to the sparse matrix represented with CSR with a similar pruning rate. This is because VCM's index storage is reduced by **85.2%** compared to CSR's index size. Even if the CSR is represented with relative index using 5 bits (Han et al., 2016b), at the expense of increased index decoding complexity, the VCM index size is still smaller by **52.7%**[6].

In summary, VCM is superior to CSR due to its encoded index format that requires a smaller storage requirement and parallel dense matrix reconstruction process through VD while maintaining a comparable model accuracy.

## 3.2 ALEXNET ON IMAGENET RESULTS

We verified the scalability of the VCM and Viterbi-based pruning methods using the AlexNet model on ImageNet. The number of VD outputs is 50 for both the FC1 and FC2 layers ($NUM_v$=50, $NUM_c$=5, $R$=10) and 8 for the other layers ($NUM_v$=8, $NUM_c$=4, $R$=2). Similar to the MNIST results, a higher index compression ratio is set for layers with larger number of weights. Since the skip state is one, the index compression ratio becomes $R/2$. The minimum Hamming distance and the XOR taps are 4. Table 3 presents the pruning rates and matrix sizes assuming that non-zero weights and CSR index are stored with 16-bit format.

The **38.1%** reduction in matrix size achieved using VCM is mainly due to the significant reduction in the index storage requirement (**83.9%**). Compared with the 4-bit relative index scheme introduced in (Han et al., 2016b), the index size of VCM is reduced by **35.5%**. The advantage of the index compression ratio of the proposed technique is largely attributed to the VD's limited search space out of all possible encodable index formats, while pruning methods employing traditional sparse matrix formats do not consider such restriction. Despite such limitation, both methods achieve similar top-1 and top-5 classification accuracy with the same retraining time.

---

[6]Additional size reductions techniques, such as quantizing non-zero weights and Huffman coding (Han et al., 2016b), can also be applied to our methods

## 4 RELATED WORK

Denil et al. (2013) demonstrated that most neural networks parameters have significant redundancy. The redundancy increases the system complexity, and causes overfitting with small training dataset. Several approaches have been suggested to prune deep neural networks and increase the sparsity of parameters in order to minimize both the memory overhead and the computation time, and avoid overfitting.

Chauvin (1989) and Hanson & Pratt (1989) introduced additional cost biases to the objective function to decay the unimportant parameters. LeCun et al. (1990) and Hassibi et al. (1993) suggested pruning parameters while minimizing the increase of error approximated by Hessian matrix. Optimal Brain Damage (OBD) (LeCun et al., 1990) restricts the Hessian matrix, forcing it to be diagonal to reduce the computational burden, at the cost of additional performance degradation. Optimal Brain Surgeon (OBS) (Hassibi et al., 1993) used a full Hessian matrix with additional computation cost to improve the pruning performance.

Han et al. (2015) proposed pruning of deep neural networks by removing parameters based on the magnitude of their absolute values and then iteratively retraining the pruned network. A $9\times$ and $13\times$ pruning rate was achieved for AlexNet and VGG-16, respectively, without loss of accuracy on ImageNet dataset. A follow-up paper compressed the pruned network further with weight sharing and Huffman coding (Han et al., 2016b). Although an impressive compression rate is achieved by these suggested methods, the irregular sparsity of the survived parameters and the associated complicated index decoding process prevent common hardware such as GPUs from achieving noticeable speed-up improvement. Alternatively, Han et al. (2016a) designed a dedicated hardware accelerator to circumvent this problem.

Recently, several papers suggested iterative hardware-efficient pruning methods to realize a faster inference speed and smaller model size. Molchanov et al. (2017c) suggested iterative pruning on a feature-map level based on a heuristic approach to evaluate the importance of parameters. This paper, which shares a similar idea as that of OBS, uses first-degree Taylor polynomial to estimate the importance of each parameter with reduced computational burden. Since the method prunes feature maps rather than each parameter, a sparse matrix format is not required at the cost of a lower pruning rate. Li et al. (2017) suggested pruning all the convolution kernels together with corresponding feature maps in CNN. Similar to Molchanov et al. (2017c), this coarse-level pruning avoids the use of a sparse matrix format, at the expense of a lower pruning rate. Park et al. (2017) introduced a high-performance sparse convolution algorithm, where the sparse convolution was formulated as sparse-matrix-dense-matrix multiplication with the dense matrix generated on the fly. The paper shows that this method can improve the inference speed of pruned networks with moderate sparsity, and can prune each parameter independently, leading to a better pruning rate. However, in the paper, the results were only demonstrated on CPUs; it was not shown whether the proposed method can also be applied to throughput-oriented hardware such as GPUs.

Ardakani et al. (2017) proposed a scheme to generate a masking matrix using linear-feedback shift registers (LFSRs) to randomly prune some of the synaptic weights connections. Even though the hardware structure for pruning can be simplified, it is not possible to selectively prune connections to improve the pruning quality. In addition, the scheme can only be applied to the fully-connected layer, not to the convolution layer.

Kingma et al. (2015) explained Gaussian Dropout as a special case of Bayesian regularization. Unlike Gaussian Dropout which considers dropout rates as a hyperparameter, Variational Dropout theoretically allows training dropout rates layer-wise, or even weight-wise. However, the paper did not include any experimental result on weight-wise variational dropout. Molchanov et al. (2017a) extended Kingma et al. (2015) and showed the working case of weight-wise Variational Dropout. Molchanov et al. (2017a) suggested the use of this characteristic of Variational Dropout to prune deep neural networks. By pruning out weights with a high dropout rate, high sparsity on a deep neural network was achieved for the CIFAR-10 classification task. Molchanov et al. (2017b) and Louizos et al. (2017) suggested pruning deep neural networks in a structured format with new Bayesian models. The papers could prune deep neural networks either neuron-wise or channel-wise, keeping the weight matrices in dense format. Both papers showed state-of-the-art sparsity on deep neural networks for the CIFAR-10 classification task.

In multiple works, attempts have been made to reduce the redundancy with popular lossy compression methods. Denton et al. (2014) applies low rank approximations to pre-trained weights. Gong et al. (2014) uses vector quantization to compress deep convolution neural networks. Chen et al. (2015) suggests HashedNets, which applies hashing tricks to reduce the model sizes. Iandola et al. (2016) achieves AlexNet-level accuracy using 50x fewer parameters with SqueezeNet, which is comprised of custom convolution filters called Fire modules. These methods are orthogonal to the network pruning, and can be combined to achieve further model compression. For example, SqueezeNet combined with Deep Compression (Han et al., 2016b) achieves $510\times$ compression ratio compared to the original AlexNet.

## 5 FUTURE WORK

Many other ECC techniques have been reported that can also be potentially used to search for sparse matrix forms with high index compression (Morelos-Zaragoza, 2006). Efficient parallel ECC decoding and encoding implementation have also been proposed and realized (Zhang, 2015). We believe that efforts to combine existing and new ECC techniques/algorithms with DNN pruning methods create a new dimension in realizing energy-efficient and high-performance DNN. Even though the proposed approach is best for dedicated ASIC or FPGA, the inherent parallel characteristics of VD and the Viterbi algorithm can also be utilized in GPUs through the construction of new kernels and libraries. We have not considered quantization of non-zero weight values or entropy-related coding design in this paper. In the future, such considerations can be embedded into the branch metric or path metric equations.

## 6 CONCLUSION

We proposed a new DNN-dedicated sparse matrix format and pruning method using the Viterbi encoder structure and Viterbi algorithm. Unlike previous methods, we first consider only limited choices of pruning results, all of which have the advantage of a significant index compression ratio by our proposed index decompressing structures. One particular pruning result is selected from the limited pruning solution space based on the Viterbi algorithm with user-defined branch metric equations that aim to minimize the accuracy degradation. As a result, our proposed sparse matrix, VCM, shows noticeable index storage reduction even compared with the relative index scheme. Fixed index compression ratio and inherently parallel reconstruction scheme allows a wide range of applications, such as SpMM, since sparse matrices can be converted into dense matrices efficiently.

### ACKNOWLEDGMENTS

This research was in part supported by the MSIT(Ministry of Science and ICT), Korea, under the ICT Consilience Creative program(IITP-2017-R0346-16-1007) supervised by the IITP(Institute for Information & Communications Technology Promotion)

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

# A  APPENDIX

## A.1  VD DESIGN METHODOLOGY

In Figure 1, each VD output is generated by a series of 2-input XOR gates which accept input bits from either input of VD or FF outputs. Hence, there are 6 possible input candidates in total for XOR gates and each candidate is called an XOR tap. Using input as $x^0$ and $n^{th}$ FF output (from the left) as $x^n$, $out2$ can be represented as a polynomial of $x^5 + x^3 + x$ or equivalently, a vector $[101010]$. By combining such vectors of all 4 outputs, we can construct a VD Matrix to represent VD (of Figure 1) in a compact manner as the following:

$$\begin{bmatrix} 1 & 1 & 0 & 1 & 0 & 0 \\ 1 & 0 & 1 & 0 & 1 & 0 \\ 0 & 1 & 1 & 0 & 0 & 1 \\ 0 & 0 & 0 & 1 & 1 & 1 \end{bmatrix}. \tag{3}$$

The number of 1s (i.e., the XOR taps) is 3 in every row of VD Matrix and the Hamming distance[7] of any pair of two rows is 4. Increasing the number of XOR taps and minimum Hamming distance in VD Matrix improves the randomness of VD outputs (Lee & Roy, 2012). Given the number of XOR taps, the minimum Hamming distance, and the number of VD outputs, VD Matrix can be generated as Algorithm 1.

---

**Algorithm 1:** VD Matrix generation

---

**input** : number of outputs $N$, number of XOR taps $t$,
        minimum Hamming distance $h$
**output:** VD Matrix $S$
$i = 0$, $S = \phi$ ;
**while** *(number of vectors of S)* $< N$ **do**
     $i$++ ;
     $a$ = binary representation of $i$ ;
     **if** *(number of 1s' of a)* $== t$ **then**
         isValid = **true** ;
         **for** *(j=0; j <number of vectors of S; j++)* **do**
             $d$ = Hamming distance between $S(j)$ and $a$;
             **if** $d < h$ **then**
                 isValid = **false** ;
             **end**
         **end**
         **if** *(isValid* $==$ *true)* **then**
             put $a$ in $S$;
         **end**
     **end**
**end**

---

Table 4 shows the minimum number of FFs generated by Algorithm 1, given the number of XOR taps, the minimum Hamming distance, and the number of VD outputs. Note that the number of VD outputs increases exponentially, while the number of FFs increases linearly. Thus, the hardware resource for implementing VD is not expensive even with a high index compression ratio. Note that the number of XOR taps for pruning should always be an even number (otherwise, the Viterbi algorithm chooses a trivial input sequence of all '1's to ensure make all the weights survive to maximize the path metric.

In Figure 10, as further bits are consumed for the two inputs of the comparators ($NUM_c$ bits) in order to enhance controllable target pruning rates (i.e., sparsity in $R$ outputs) resolution, the number of VD outputs ($NUM_v$) needs to be increased.

---

[7]Hamming distance between two vectors is the number of positions where two values are different

Table 4: Various configurations of the number of XOR taps, the minimum Hamming distance, the number of FFs, and the number of VD outputs.

| # of VD outputs | # of taps | Hamming | # of FFs |
|---|---|---|---|
| 8 | 5 | 2 | 6 |
| | 5 | 4 | 8 |
| | 5 | 6 | 10 |
| | 4 | 6 | 12 |
| 32 | 4 | 2 | 6 |
| | 5 | 4 | 10 |
| | 5 | 6 | 15 |
| | 6 | 8 | 18 |
| 128 | 6 | 2 | 9 |
| | 7 | 4 | 13 |
| | 8 | 6 | 17 |
| | 6 | 6 | 19 |

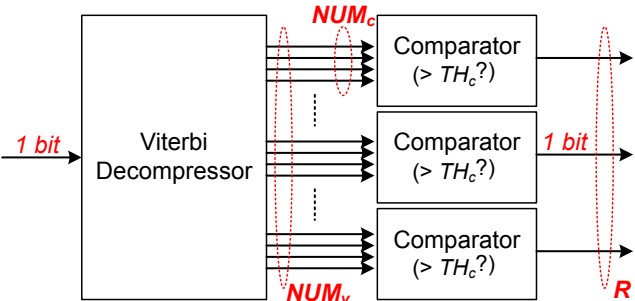

Figure 10: Index decompressing using VD and comparators to control the sparsity. A comparator threshold value $TH_c$ can have a range of 0 to $2^{NUM_c} - 1$.

## A.2 DUMMY INPUTS AND SKIP STATES

Accuracy degradation can be reduced by increasing the number of states to be explored in trellis diagram, primarily because of the increased search space dimension. In Figure 11, inserting dummy inputs as an initial input sequence increases the number of available states from which we start index encoding with weight parameters (i.e., the number of available states increases from 1 to 4 with 2 dummy input bits). The maximum size of the dummy inputs is the number of FFs and all the dummy paths exhibit the same preference with the same branch metrics. The size of the dummy inputs is negligible if the number of FFs in VD is much smaller than the number of weight matrix elements divided by $R$.

In addition to the dummy input sequence, the skip state, which is defined as the number of times the time index in the trellis diagram is skipped, can also lead to reduced accuracy degradation. Similar to the concept of the dummy input, the skip state increases the number of available states in the trellis diagram search. Figure 12 describes a case of (Skip State=1) where at every even-number time index, the output of VD is discarded. If the branch metrics are set to 0, following Eq. (1), the Viterbi algorithm will select the paths that lead to an increased number of states with a higher path metric value. In the case of magnitude-based pruning, this implies that a larger magnitude weight has a higher chance of being preserved. In the case of the $k$ skip states, VD outputs are discarded for $k$ consecutive time indices. The entire length of the time index is increased by $(k + 1)$ times and the index compression ratio is reduced by $(k + 1)$ times.

## A.3 EXAMPLE OF SELECTING FUNCTIONS FOR THE BRANCH METRIC

Assume that the normalized magnitudes of parameters before pruning are given as $\{0.1, 0.2, 0.6, 1.0\}$ at a certain time index and $TH_p$ is 0.3. We calculate the branch met-

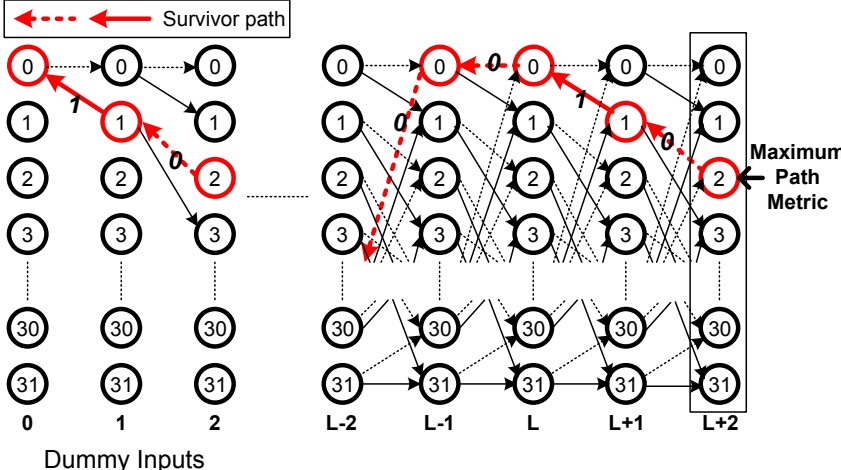

Figure 11: Backward survivor path finding procedure with 2 dummy input bits to increase the number of reachable states from 1 to 4.

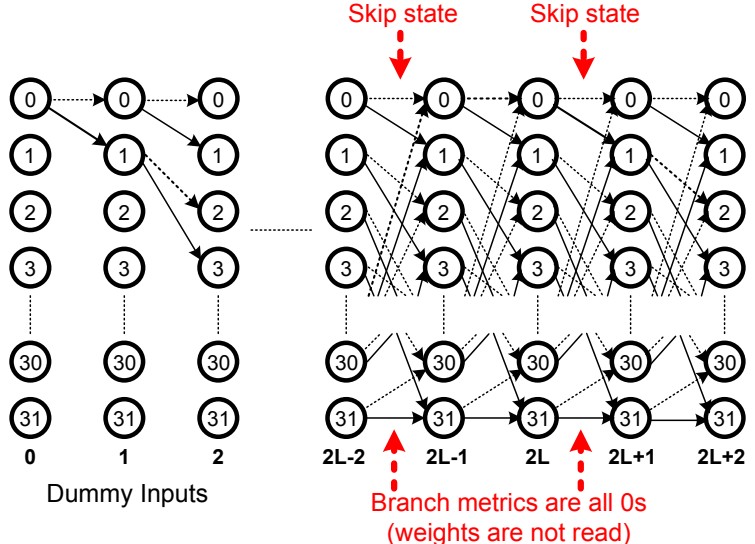

Figure 12: Trellis diagram exploration with (skip state=1). For every even-number time index after dummy input sequence, branch metrics are 0 while the path metrics are still updated. The entire time index length is increased by (skip state + 1) times.

rics with two different sets of comparator outputs $V1 = \{0, 1, 1, 0\}$ and $V2 = \{1, 0, 0, 1\}$ using Eq. (2). In the case of $\tanh(x)$, the branch metric with $V1$ is given as $S_2 \times (-\tanh((0.1 - 0.3)/S_1) + \tanh((0.2 - 0.3)/S_1) + \tanh((0.6 - 0.3)/S_1) - \tanh((1.0 - 0.3)/S_1)) = 10^4 \times (-(-0.04) + (-0.02) + (0.06) - (0.20)) = -1200$, while the branch metric with $V2$ is $10^4 \times ((-0.04) - (-0.02) - (0.06) + (0.20)) = 1200$. On the other hand, if we use $x$ instead of $\tanh(x)$, then the branch metrics with $V1$ and $V2$ are $-600$ and $600$, respectively. Compared with $x$, $\tanh$ assigns greater importance to parameters with higher magnitude. Note that in this example, $\tanh$ has been chosen in the context of the magnitude-based pruning method, and the branch metric equations can be made differently depending on the underlying pruning principle.

Table 5: Weight size and VD parameters per layer for Viterbi-based pruning.

| Network | Weight size per layer | | VD parameters | | | |
|---|---|---|---|---|---|---|
| | Original | Pruning neurons | $NUM_v$ | $NUM_c$ | R | Skip state |
| LeNet-300-100 | $784 \times 300$ | $465 \times 96$ | 40 | 5 | 8 | 1 |
| | $300 \times 100$ | $96 \times 100$ | 20 | 5 | 4 | 1 |
| | $100 \times 10$ | $100 \times 10$ | 10 | 5 | 2 | 1 |
| LeNet-5-Caffe | $5 \times 5 \times 1 \times 20$ | $5 \times 5 \times 1 \times 20$ | 10 | 5 | 2 | 1 |
| | $5 \times 5 \times 20 \times 50$ | $5 \times 5 \times 20 \times 25$ | 30 | 5 | 6 | 1 |
| | $800 \times 500$ | $400 \times 69$ | 40 | 5 | 8 | 1 |
| | $500 \times 10$ | $69 \times 10$ | 10 | 5 | 2 | 1 |

Table 6: Sparse matrix comparison with LeNet-300-100 on MNIST using Variational Dropout-based pruning (Molchanov et al., 2017a) and our proposed Viterbi-based pruning. We assume that non-zero values and CSR index use 16 bits.

| Layer | Weight Size | Variational Dropout | | Additional Viterbi-Pruning | | Matrix Size Reduction |
|---|---|---|---|---|---|---|
| | | Pruning Rate | Sparse Matrix (CSR) Size | Pruning Rate | Sparse Matrix (VCM) Size | |
| FC1 | 89.28K | 95.7% | 7.67KB | 94.8% | 5.89KB | 23.2% |
| FC2 | 19.2K | 94.6% | 2.23KB | 94.0% | 1.71KB | 23.6% |
| FC3 | 20.0K | 78.0% | 0.88KB | 68.0% | 0.75KB | 15.2% |
| Total | 128.48K | **95.2%** | 10.78KB | **94.2%** | 8.34KB | **22.6%** |
| Test Error | | 1.96% | | 1.96% | | |

Table 7: Sparse matrix comparison with LeNet-5-Caffe on MNIST using Variational Dropout-based pruning (Molchanov et al., 2017a) and our proposed Viterbi-based pruning. We assume that non-zero values and CSR index use 16 bits.

| Layer | Weight Size | Variational Dropout | | Additional Viterbi-Pruning | | Matrix Size Reduction |
|---|---|---|---|---|---|---|
| | | Pruning Rate | Sparse Matrix (CSR) Size | Pruning Rate | Sparse Matrix (VCM) Size | |
| Conv1 | 1.0K | 69.6% | 0.63KB | 69.6% | 0.36KB | 43.6% |
| Conv2 | 25.0K | 96.4% | 1.80KB | 95.8% | 1.52KB | 15.4% |
| FC1 | 55.2K | 97.5% | 2.83KB | 97.0% | 2.49KB | 12.2% |
| FC2 | 1.38K | 66.1% | 0.94KB | 66.7% | 0.53KB | 43.0% |
| Total | 82.58K | **96.7%** | 6.20KB | **96.1%** | 4.90KB | **21.0%** |
| Test Error | | 0.96% | | 0.98% | | |

## A.4 VITERBI ALGORITHM ON HIGHLY SPARSE DNNS

We test our proposed VCM and Viterbi-based pruning method with highly sparse DNNs by combining our method with the Variational dropout-based pruning method (Molchanov et al., 2017a). The LeNet-300-100 and LeNet-5-Caffe architectures[8] on the MNIST dataset are used in the test. The Viterbi-based pruning based in Eq. (2) is additionally performed to obtain VCM data, after pruning weights and neurons using the Variational dropout method (Molchanov et al., 2017a) as shown in Table 5. The main VD parameters for each layer are also presented in Table 5. The minimum Hamming distance and XOR taps are 4. Table 6 and 7 describe the pruning rate and the memory footprint comparisons for each layer. The storage requirement for VCM is reduced by **22.6 %** and **21.0 %**, respectively, compared with CSR, due to the reduction in index size (**63.0 %** and **53.6 %**, respectively). Almost the same classification accuracy is achieved with a short retraining time in both LeNet-300-100 and LeNet-5-Caffe. Note that this result demonstrates that the proposed Viterbi-based techniques can be combined with existing pruning methods to extract VCM formats even without modifying Eq. (2).

---

[8]https://github.com/ars-ashuha/variational-dropout-sparsifies-dnn

