# OpenReview forum: "Viterbi-based Pruning for Sparse Matrix with Fixed and High Index Compression Ratio"
_ICLR.cc/2018/Conference — Accept (Poster)_

### Official Review · AnonReviewer1 · 2017-11-25
**Solid work but question its usefulness**

**Rating:** 6
**Confidence:** 4

**Review:**

It seems like the authors have carefully thought about this problem, and have come up with some elegant solutions, but I am not sold on whether it's an appropriate match for this conference, mainly because it's not clear how many machine learning people will be interested in this approach.

There was a time about 2 or 3 years ago when sparse-matrix approaches seemed to have a lot of promise, but I get the impression that a lot of people have moved on.  The issue is that it's hard to construct a scenario where it makes sense from a speed or memory standpoint to do this.  The authors seem to have found a way to substantially compress the indexes, but it's not clear to me that this really ends up solving any practical problem.  Towards the end of the paper I see mention of a 38.1% reduction in matrix size.  That is way too little to make sense in any practical application, especially when you consider the overhead of decompression.   It seems to me that you could easily get a factor of 4 to 8 of compression just by finding a suitable way to encode the floating-point numbers in many fewer bits (since the weight parameters are quite Gaussian-distributed and don't need to be that accurate).

---

> ### Author Response · Authors · 2017-12-27
> **Response to AnonReviewer1**
>
> We thank the AnonReviewer#1 for taking the time to review this paper.
>
> We agree with the reviewer that quantization is another effective scheme to reduce memory footprint. We, however, believe that the pruning and quantization are two orthogonal approaches that both aim to help reducing the memory/computation overhead as demonstrated in S. Han et al.’s Deep Compression paper, for example. To answer the reviewer’s question in more detail, we also experimented quantization of weight values for the viterbi-pruned LeNet-5 on MNIST dataset. Similar to S. Han et al.’s case, we observed that 4-bit weight quantization could maintain the inference accuracy, which confirms that pruning and quantization can be applied together and further reduce the memory footprint.
>
> In addition, recent works, such as variational dropout (as discussed by reviewer #2), show that it is possible to prune and compress a large model(i.e., vgglike) by more than 50x without loss of accuracy. The results indicate that sparse-matrix approaches still have merits for further investigation. As we have shown in response to reviewer #2, the VCM format can also work with this approach (in fact, we believe VCM can work with any underlying pruning approach).
>
> Regarding the reviewer’s concern for VCM decompression overhead, we believe that the overhead is very small because it requires small number of FFs and XOR gates as shown in the Figure 1 and Table 4. Please note that VCM decompression corresponds to the Viterbi encoding in error correction code applications, which is regarded as a much lighter process compared to Viterbi decoding.

---

### Official Review · AnonReviewer2 · 2017-11-26
**Viterbi-based pruning review**

**Rating:** 6
**Confidence:** 4

**Review:**

The paper proposes VCM, a novel way to store sparse matrices that is based on the Viterbi Decompressor. Only a subset of sparse matrices can be represented in the VCM format, however, unlike CSR format, it allows for faster parallel decoding and requires much less index space. The authors also propose a novel method of pruning of neural network that constructs an (sub)optimal (w.r.t. a weight magnitude based loss) Viterbi-compressed matrix given the weights of a pretrained DNN.
VCM is an interesting analog to the conventional CSR format that may be more computationally efficient given particular software and/or hardware implementations of the Viterbi Decompressor. However, the empirical study of possible acceleration remains as an open question.
However, I have a major concern regarding the efficiency of the pruning procedure. Authors report practically the same level of sparsity, as the pruning procedure from the Deep Compression paper. Both the proposed Viterbi-based pruning, and Deep Compression pruning belong to the previous era of pruning methods. They separate the pruning procedure and the training procedure, so that the model is not trained end-to-end. However, during the last two years a lot of new adaptive pruning methods have been developed, e.g. Dynamic Network Surgery, Soft Weight Sharing, and Sparse Variational DropOut. All of them in some sense incorporate the pruning procedure into the training procedure and achieve a much higher level of sparsity (e.g. DC achieves ~13x compression of LeNet5, and SVDO achieves ~280x compression of the same network). Therefore the reported 35-50% compression of the index storage is not very significant.
It is not clear whether it is possible to take a very sparse matrix and transform it into the VCM format without a high accuracy degradation. It is also not clear whether the VCM format would be efficient for storage of extremely sparse matrices, as they would likely be more sensitive to the mismatch of the original sparsity mask, and the best possible VCM sparsity mask. Therefore I’m concerned whether it would be possible to achieve a close-to-SotA level of compression using this method, and it is not yet clear whether this method can be used for practical acceleration or not.
The paper presents an interesting idea that potentially has useful applications, however the experiments are not convincing enough.

---

> ### Author Response · Authors · 2017-12-27
> **Response to AnonReviewer2**
>
> Thank you very much for the constructive comments. We added Appendix A.4 to show that SVDO can be combined with our proposed VCM format.
>
> We believe that our proposed technique is a general one and can be combined with SVDO since our proposed Viterbi encoder/algorithm is basically not dedicated to specific pruning methods (rather, we wanted to suggest a new sparse matrix format which is better suited to DNN). As shown in the Table 6 and 7 in Appendix A.4, for MNIST dataset, we could achieve a competitive pruning rate and memory footprint reduction by applying our proposed Viterbi algorithm to the sparse matrix produced from SVDO scheme after removing fully disconnected neurons. We hope Appendix A.4 addresses your concern that VCM may not be able to handle such highly sparse matrices.
>
> In the original draft, one of our main experimental study was to show pruning results for a large benchmark such as ImageNet database (or something of a similar scale) and please note that we still compare the pruning results with S. Han et al.’s method for ImageNet as SVDO did not report ImageNet results.

---

### Official Review · AnonReviewer3 · 2017-11-27
**The authors use Viterbi encoding to dramatically compress the sparse matrix index of a pruned network, reducing one of the main memory overheads of a pruned neural network and speeding up inference in the parallel setting.**

**Rating:** 7
**Confidence:** 3

**Review:**

quality: this paper is of good quality
clarity: this paper is very clear but contains a few minor typos/grammatical mistakes (missing -s for plurals, etc.)
originality: this paper is original
significance: this paper is significant

PROS
- Using ECC theory for reducing the memory footprint of a neural network seems both intuitive and innovative, while being grounded in well-understood theory.
- The authors address a consequence of current approaches to neural network pruning, i.e., the high cost of sparse matrix index storage.
- The results are extensive and convincing.

CONS
- The authors mention in the introduction that this encoding can speed up inference by allowing efficient parallel sparse-to-dense matrix conversion, and hence batch inference, but do not provide any experimental confirmation.

Main questions
- It is not immediately clear to me why the objective function (2) correlates to a good accuracy of the pruned network. Did you try out other functions before settling on this one, or is there a larger reason for which (2) is a logical choice?
- On a related note, I would find a plot of the final objective value assigned to a pruning scheme compared to the true network accuracy very helpful in understanding how these two correlate.
- Could this approach be generalized to RNNs?
- How long does the Viterbi pruning algorithm take, as it explores all 2^p possible prunings?
- How difficult is it to tune the pruning algorithm hyper-parameters?

---

> ### Author Response · Authors · 2017-12-27
> **Response to AnonReviewer3**
>
> We thank you for your feedback and comments. We address your concerns and questions below.
>
> “It is not immediately clear to me why the objective function (2) correlates to a good accuracy of the pruned network. Did you try out other functions before settling on this one, or is there a larger reason for which (2) is a logical choice?”
>
> We tried various objective functions such as x, x^2, exp(x), tanh(x) and sigmoid(x) as shown in Fig. 9. While all of the objective functions gave comparable accuracy, we chose tanh(x) because, in the context of magnitude-based pruning, selecting a few parameters with large magnitude is more desirable than choosing many parameters with medium magnitude, even though the sum of the (survived parameters’) magnitude can be the same in both cases. Hence, in order to obtain a highly skewed distribution of survived parameters, ‘tanh’ is considered in our experiments. We added Appendix A.3 to show how ‘tanh’ and ‘x’ can produce different branch metric values, given a list of parameters and comparator outputs. Please note that there can be many other objective functions that potentially result in similar/better skewed distributions.
>
> “I would find a plot of the final objective value assigned to a pruning scheme compared to the true network accuracy very helpful in in understanding how these two correlate.”
>
> We hope our response above addresses this concern as well. Since the Viterbi algorithm finds an optimal sequence maximizing the path metric, there is no immediate relationship between the branch metric (i.e., Eq.(2)) and the network accuracy. However, producing high branch metrics would increase the chance to improve the final path metric and to select parameters with large magnitude after a sequence-level optimization.
> At the final time index, all path metrics share a long optimal sequence through the survivor selection procedure (as a result, it is not necessary to investigate 2^p paths). Choosing any path metric at the final time index, hence, would produce almost the same accuracy. Please note that comparing absolute values of the final path metrics with different branch metric equations would be meaningless due to the path metric normalization.
>
> “Could this approach be generalized to RNNs?”
>
> Yes, our proposed approach can be generalized to RNN for which a pruning is applicable. For instance, we performed an experiment with a language model on PTB dataset (medium size, https://github.com/tensorflow/models/tree/master/tutorials/rnn/ptb). The magnitude-based pruning and the Viterbi-based pruning obtain 81.4% and 81.5% pruning rates, respectively, while VCM has a storage reduction of 43.9% compared with CSR and the perplexity degradation is comparable between the two cases. Since there are no widely accepted benchmarks for RNN pruning experiments, we did not include this experimental result in the manuscript.
>
> “How long does the Viterbi algorithm take, as it explores all 2^p possible prunings?”
>
> Due to the dynamic programming property of the Viterbi algorithm, it is not necessary to compute all 2^p possible prunings. The time complexity of the Viterbi algorithm is linearly proportional to the number of states in the trellis diagram and p (not 2^p). There are many well-known implementation techniques to reduce the time complexity of the Viterbi algorithm, such as a sliding window technique (e.g., the time complexity becomes independent of the length of the input sequence), which could not be included in our manuscript due to the space limit.
>
> “How difficult is it to tune the pruning algorithm hyper-parameters?”
>
> In our manuscript, where a magnitude-based pruning is used as a baseline, the difficulty of tuning hyper-parameters using our proposed methodology is almost the same as that of the magnitude-based pruning, because other than VH_p, selecting hyper-parameters becomes trivial. For example, increasing NUM_c, XOR taps, and the Hamming distance always improves the pruning rate and the compression rate (while typical numbers for those hyper-parameters introduced in the paper are good enough). The comparator threshold (TH_c) is automatically determined by the target pruning rate, which is dominated by TH_p. Finding an appropriate VH_p value follows the way of magnitude-based pruning methods.

---

### Decision · Program_Chairs · 2018-01-29
**ICLR 2018 Conference Acceptance Decision**

**Decision:**

Accept (Poster)

**Comment:**

The paper proposes a new sparse matrix representation based on Viterbi algorithm with high and fixed index compression ratio regardless of the pruning rate.  The method allows for faster parallel decoding and achieves improved compression of index data storage requirement over existing methods (e.g., magnitude-based pruning) while maintaining the pruning rate. The quality of paper seems solid and of interest to a subset of the ICLR audience.